# Regional Analysis of Associations between Infant and Young Child Feeding Practices and Diarrhoea in Indian Children

**DOI:** 10.3390/ijerph17134740

**Published:** 2020-07-01

**Authors:** Mansi Vijaybhai Dhami, Felix Akpojene Ogbo, Thierno M.O. Diallo, Kingsley E. Agho

**Affiliations:** 1Translational Health Research Institute (THRI), School of Medicine, Western Sydney University, Campbelltown Campus, Penrith, NSW 2571, Australia; f.ogbo@westernsydney.edu.au (F.A.O.); k.agho@westernsydney.edu.au (K.E.A.); 2General Practice Unit, Prescot Specialist Medical Centre, Welfare Quarters, Makurdi, Benue State 972261, Nigeria; 3School of Social Sciences, Western Sydney University, Penrith Campus, Penrith, NSW 2571, Australia; t.diallo@westernsydney.edu.au; 4Statistiques & M. N., Sherbrooke, QC J1K 2Z4, Canada; 5African Vision Research Institute (AVRI), University of KwaZulu-Natal, Durban 4041, South Africa; 6School of Health Sciences, Western Sydney University, Campbelltown Campus, Penrith, NSW 2571, Australia

**Keywords:** diarrhoea, India, infants, young children, breastfeeding, complementary feeding

## Abstract

Studies on the association between infant and young child feeding (IYCF) practices and diarrhoea across regional India are limited. Hence, we examined the association between IYCF practices and diarrhoea in regional India. A weighted sample of 90,596 (North = 11,200, South = 16,469, East = 23,317, West = 11,512, Central = 24,870 and North-East = 3228) from the 2015–2016 National Family Health Survey in India was examined, using multivariate logistic regressions that adjust for clustering and sampling weights. The IYCF indicators included early initiation of breastfeeding (EIBF), exclusive breastfeeding (ExcBF), predominant breastfeeding (PBF), bottle feeding (BotF), continued breastfeeding (BF) at one-year, continued BF at two years, children ever breastfed and the introduction of solid, semi-solid or soft foods (ISSSF). Diarrhoea prevalence was lower among infants who were BF within one-hour of birth and those who were exclusively breastfed. Multivariate analyses revealed that continued BF at one and two years, and infants who were introduced to complementary foods had a higher prevalence of diarrhoea. EIBF and ExcBF were protective against diarrhoea in the regions of North, East and Central India. PBF, BotF and ISSSF were risk factors for diarrhoea in Central India. Continued BF at two years was a risk factor for diarrhoea in Western India. Findings suggested that EIBF and ExcBF were protective against diarrhoea in Northern, Eastern and Central India, while PBF, BotF, continued BF at two years and ISSSF were risk factors for diarrhoea in various regions in India. Improvements in IYCF practices are likely to reduce the burden of diarrhoea-related morbidity and mortality across regions in India.

## 1. Introduction

Appropriate infant and young child feeding (IYCF) practices can reduce childhood infections (such as diarrhoea) and provide a strong foundation for the optimal growth and development of children [1]. In low- and middle-income countries (LMICs), diarrhoea is still a leading cause of death and health loss among children younger than five years. A recent study has indicated that inappropriate IYCF substantially contributed to the 1.3 million global diarrhoea-related deaths and 1.9 million health loss [2]. In India, inappropriate IYCF practices remain prevalent. For example, recent studies have indicated that the prevalence of early initiation of breastfeeding (EIBF) was 41% [3] and that for exclusive breastfeeding (ExcBF) was 55%, with substantial differences within and between regional areas [4]. Moreover, the proportion of Indian women who appropriately introduced solid, semi-solid or soft foods to their infants ranged from 38% in both North and Central India to 61% in Southern India [5]. These inappropriate IYCF practices may have considerable adverse implications for diarrhoeal disease burden among Indian infants and young children.

In recent years, India has implemented a range of interventions to reduce diarrhoeal disease burden. These interventions included the Integrated Child Development Scheme; Water, Sanitation and Hygiene (WASH) trials [6]; National Diarrhoeal Disease Control Programme [7,8]; and policies for childhood nutritional development [9]. However, in 2016, India accounted for a substantial proportion of diarrhoea-related deaths in South Asia (65%) [2]. Additionally, various studies have shown wide differences in the prevalence of diarrhoea across Indian regions which ranged from 23% in Southern India [10] and 25% in Kashmir [11] to 39% in Western Maharashtra [12]. Most of the diarrhoeal-related deaths may be attributable to a lack of potable water [13], poor hygiene and sanitation [14], and low immunisation coverage [15], as well as inappropriate IYCF behaviours [16].

In India, some discrete studies have highlighted the impact of IYCF on diarrhoeal disease among children. Studies conducted in West Bengal [17,18], Maharashtra [12] and Southern India [19] had indicated that inappropriate breastfeeding was associated with the onset of diarrhoea among children under 5 years of age. A systematic review suggested that the incidence of childhood diarrhoeal disease may be due to a range of factors, including malnutrition and suboptimal breastfeeding [16]. However, the review did not define what was meant by suboptimal breastfeeding, nor did the study highlight to what extent (in terms of effect size) the specific IYCF indicators were related to diarrhoea in regional India. The previous studies did not use data from the most recent National Family Health Survey (NFHS-4) to provide a detailed assessment of the association between IYCF practices and diarrhoea at the regional level of India. The new data are likely to reflect the current demographic, socioeconomic and health service situation in India.

Comprehensive subnational assessment of the relationship between IYCF behaviours and diarrhoea is essential to stakeholders in India, as national data can often mask important differences across regional areas [4]. Furthermore, a regional analysis of the association between IYCF practices and diarrhoea in India is needed to better understand to what extent each of the relevant IYCF behaviour act as a “protector” against or “predictor” for the onset of diarrhoea among infants and young children. This information is relevant to region-specific health practitioners, policy makers and breastfeeding advocates, given the significant disparities in socioeconomic and health service indicators across India [20,21,22,23]. Accordingly, this study investigated the association between IYCF practices and diarrhoeal disease among infants and young children at the regional level in India.

## 2. Method

### 2.1. Data Sources

The study used data from the 2015–2016 India Demographic and Health Survey (also called the National Family Health Survey, NFHS-4), conducted by International Institute for Population Sciences, Mumbai, through the Ministry of Health and Family Welfare (MoHFW), Government of India. Data collection was technically supported by the Inner City Fund (ICF) International, Maryland, USA. Information on sociodemographic and household characteristics and IYCF practices was collected from a nationally representative sample of eligible women aged 15–49 years. Eligible women were either permanent residents of the surveyed households or visitors who stayed in the households the night before the survey. The response rate in the survey ranged from 94.0% (Andhra Pradesh or West Bengal) to 99.6% in Bihar [24].

In the NFHS-4, a total sample of 249,454,252 households were surveyed based on the 2011 census frame. The households were selected by using a two-staged sampling design both for rural and urban areas. Within each rural area, the probability proportional to size was used initially to select villages from a sampling frame and were designated as the Primary Sampling Units (PSUs). The second stage involved the random selection of households from each PSU. In the urban areas, Census Enumeration Blocks (CEBs) were selected in the first stage. The second stage involved the random selection of households from each CEB. Further information on the sampling methodology and data collection has been provided in the final NFHS-4 report [24]. In this study, a total weighted sample of 90,596 maternal responses was used, and the analyses were restricted to the young child aged 0–23 months, living with the respondent, to reduce the potential impact of recall bias, and this approach was consistent with previous studies [4,5].

### 2.2. Study Setting

India is a federation that comprises 29 states and 7 union territories. The states and union territories are categorised into six administrative zones, to facilitate and improve financial allocation, economic integration and inter-state cooperation [25,26]. The six zonal regions include North, South, East, West, Central and North-Eastern India. The Northern region (*n* = 11,200) consists of the states and union territories of Jammu and Kashmir, Himachal Pradesh, Haryana, Delhi, Chandigarh, Punjab and Rajasthan. The Southern region (*n* = 16,469) consists of the states and union territories of Andhra Pradesh, Karnataka, Kerala, Tamil Nadu, Telangana, Andaman and Nicobar Islands, Lakshadweep Islands and the Union Territory of Puducherry. The Eastern region (*n* = 23,317) consists of the states of Bihar, Jharkhand, Odisha and West Bengal. The Western region (*n* = 11,512) consists of the states and union territories of Gujarat, Maharashtra, Goa, Daman and Diu, as well as Dadra and Nagar Haveli. The Central region (*n* = 24,870) consists of the states of Chhattisgarh, Madhya Pradesh, Uttar Pradesh and Uttarakhand. The North-Eastern region (*n* = 3228) consists of the states of Arunachal Pradesh, Assam, Manipur, Meghalaya, Mizoram, Nagaland, Sikkim and Tripura. In 2019, the Indian government announced that the state of Jammu and Kashmir and Ladakh had been administratively re-organised into union territories [27]. However, we considered Jammu and Kashmir and Ladakh to be a part of the state of Jammu and Kashmir, due to the nature of the data.

### 2.3. Study Outcome 

Diarrhoea was the main outcome variable in this study, defined as the passage of three or more loose or liquid stools per day [24]. Mothers were asked whether the child under the age of five years had experienced symptoms of diarrhoea in the 2 weeks prior to the survey. In our study, the measurement of diarrhoea was specific to the child age group for each of the IYCF indicators, and this method was consistent with previously published studies [28,29,30].

### 2.4. Exposure Factors

The exposure variables were the IYCF indicators, defined in accordance with the World Health Organisation (WHO) definition for assessing IYCF practices in populations [31]. These IYCF indicators were selected based on past studies from LMICs, which showed that these indicators were associated with diarrhoea [28,29,30,32,33,34].
EIBF or timely initiation of breastfeeding was defined as the proportion of children within 0–23 months of age who were breastfed within one hour of birth.ExcBF was defined as the proportion of infants 0–5 months of age who received breast milk as the only source of nourishment but allowed oral rehydration solution, drops or syrups of vitamins and medicines.Predominant breastfeeding (PBF) was defined as the proportion of infants 0–5 months of age who received breast milk as the main source of nourishment but allowed water, water-based drinks, fruit juice, oral rehydration solution, drops or syrups of vitamins and medicines.Bottle feeding (BotF) was defined as the proportion of children 0–23 months of age who were fed with a bottle during the previous day.Continued breastfeeding (BF) at 1 year was defined as the proportion of children 12–15 months of age who were fed breast milk.Continued BF at 2 years was defined as the proportion of children 20–23 months of age who were fed breast milk.Children ever breastfed was defined as the proportion of children born in the last 24 months who were ever breastfed.Introduction of solid, semi–solid or soft foods (ISSSF) was defined as the proportion of infants 6–8 months of age who received solid, semi–solid or soft foods.

### 2.5. Potential Confounding Factors

The potential confounding factors included child, maternal, family, media, health service, environmental and community-level factors, selected based on evidence from past studies [28,29,30,35,36,37]. Child factors included sex, immunization status, birth order and perceived size of the baby at birth. Maternal factors such as age, education and literacy level, employment status, type of caste or tribe and religion were considered. Respondents’ marital status and household wealth index were considered as household characteristics. In addition, media factors such as exposure to television, newspaper and radio were also considered. Health-service factors included frequency of antenatal clinic (ANC) visit, place of delivery, delivery assistance and mode of delivery. Environmental factors included source of drinking water and sanitation. Source of drinking water and the type of toilet facilities were classified as improved and unimproved based on the taxonomy of the WHO/UNICEF Joint Monitoring programme (JMP) for estimating progress on WASH [38]. Improved sources of water were defined as a piped water into dwelling; piped water to yard/plot, public tap or standpipe; tube-well or borehole; protected dug well; and protected spring or rainwater. Meanwhile, unimproved water sources consisted of unprotected spring, unprotected dug well, cart with small tank/drum, tanker-truck, surface water or bottled water. Improved sanitation facility included a flush toilet, piped sewer system, septic tank, flush/pour flush to pit latrine, ventilated improved pit latrine (VIP), pit latrine with slab, composting toilet and a special case (i.e., flush/pour flush of excreta to a place unknown to the respondent place). Unimproved sanitation facility was defined as a flush/pour flush to elsewhere (such as street, yard/plot, open sewer or a ditch), pit latrine without slab, bucket, hanging toilet or hanging latrine, shared sanitation, no facilities, bush or field. Community-level factors included designated areas of residence as urban or rural.

### 2.6. Statistical Analysis

Preliminary analysis involved the tabulation of frequencies (and corresponding percentages) of all potential confounding factors (i.e., child, maternal, family, media, health service, environmental and community level factors) by Indian regions, and this was followed by prevalence and 95% confidence intervals (CIs) of diarrhoea by each exposure variables (i.e., EIBF, ExcBF, PBF, BotF, continued BF at one year, continued BF at two years, children ever breastfed and the ISSSF) was estimated for each of the designated geographical regions in India. We employed multivariate logistic regression, using Generalized Linear Latent and Mixed Models (GLLAMM) with a logit link and binomial family to adjust for clustering and sampling weights to investigate the association between the exposures and diarrhoea for each region in India.

A similar stage modelling employed by Dibley et al. [39] was used in this analysis. A 6-staged modelling technique was employed in the multivariable modelling. In the first stage, the community and family/household factors were entered into the baseline multivariable model, to assess their relationship with the study outcome (diarrhoea). A manual stepwise backwards elimination process was performed, and variables with *p* <0.05 were retained in the first model (Model 1). In the second stage of modelling, child factors were added to into significant variables in Model 1, and variables with *p*-values < 0.05 were retained in the second stage (Model 2). Similar modelling processes were carried out accordingly for the fourth and fifth stages, environmental and health services characteristics, respectively (Models 4 and 5), and only those variables with *p*-values < 0.05 were retained in the fifth model (Model 5). In the final stage of the analysis (Model 6), the exposure factors (IYCF indicators) were added to Model 5, and variables with a *p*-value < 0.05 were retained in the final model (Model 6).

The odds ratios (ORs) and their 95% confidence intervals (CIs) obtained from the adjusted multiple logistics model were reported as the measure of association between IYCF indicators and diarrhoea. We also estimated and will report the measure of association, using the national level data to allow for comparability of the evidence. Data analyses were performed in Stata (version 14.0, Stata Corp., College Station, TX, USA).

### 2.7. Ethics

The study used existing survey datasets that are available online by application, with all identifier information removed. The surveys were approved by the Ethics Committee of the ICF International, USA, and the Ethics Review Board at the International Institute for Population Sciences, Mumbai, India. We obtained approval from Measure DHS to download and use the data for the study.

## 3. Results

### 3.1. Characteristics of the Study Population

In the study population, most mothers resided in rural areas in the Eastern (83.7%) and Central (77.7%) India regions. Most mothers belonged to poor or middle households in the Eastern (86.9%) and Central (72.7%) India regions. Southern India had the highest number of mothers who had secondary education (80.7%) and four or more ANC visits (78.4%). Most Northern (88.3%), Southern (96.5%) and Western (92.2%) Indian women delivered their babies in the health facilities. The Northern region (67.6%) had more improved sanitation compared to other regions (Appendix A). The prevalence of all IYCF indicators among children aged 0–23 months in regional India are presented in Appendix A.

### 3.2. Prevalence of Diarrhoea by IYCF Practices among Children Aged 0–23 Months in Regional India

In all the regions of India, except for the Western India region, infants who were breastfed within the first hour of birth had a lower prevalence of diarrhoea compared to those who were not breastfed within the first hour of birth (Table 1). In North and Central India, the prevalence of diarrhoea was lower among infants aged 0–5 months who were exclusively breastfed, compared those who were not exclusively breastfed. There were no changes in the prevalence of diarrhoea among infants who were exclusively breastfed and those who were not exclusively breastfed in Southern, Western and North-Eastern India (Table 1). Lower prevalence estimates of diarrhoea were observed in infants and young children whose mothers engaged in PBF and BotF behaviours across all Indian regions. Children aged 12–15 months who continued BF at one year, and those aged 20–23 months who continued BF at two years, had a higher prevalence of diarrhoea, compared to their counterparts (Table 1). Children who were ever breastfed had a higher prevalence of diarrhoea compared to those who were not breastfed. A high proportion of infants aged 6–8 months who were introduced to solid, semi–solid or soft foods experienced diarrhoea, compared to those whose mothers delayed or introduced complementary foods earlier than 6–8 months in South and North-East India (Table 1).

### 3.3. Association between IYCF Practices and Diarrhoea in Regional India

Across all regions of India, except for the regions in Western India, timely initiation of breastfeeding was protective against diarrhoea among infants and young children aged 0–23 months (Figure 1A). Infants aged 0–5 months who were exclusively breastfed were less likely to experience diarrhoea in the regions of North, East and Central India, as well as at the national level of India, compared to their counterparts (Figure 1B). Infants aged 0–5 months who were predominantly breastfeeding were more likely to experience diarrhoea compared to their counterparts in the Central India region and at the national level of India (Figure 1C). Infants and young children aged 0–23 months who were bottle fed were more likely to experience diarrhoea compared to those who were not bottle fed in the North, East and Central India regions, as well as at the national level in India (Figure 1D). Children aged 20–23 months of age who continued BF at two years were more likely to experience diarrhoea compared to their counterparts in Western India (Figure 1F). Infants aged 6–8 months who were introduced to solid, semi-solid or soft foods were more likely to experience diarrhoea compared to those whose mothers delayed or introduced complementary foods earlier than 6-8 months in the Central India region (Figure 1H). The analyses revealed that continued BF at one year and children ever breastfed were not associated with diarrhoea in India (Figure 1E,G). Unadjusted odd ratios of the eight breastfeeding indicators examined were presented in Appendix A, and the number of missing information after adjusting for potential confounding and exposure factors are in Appendix A.

## 4. Discussion

This study showed that the prevalence of diarrhoea was lower among infants and young children aged 0–23 months who were breastfed within the first hour of birth and those who were exclusively breastfed in Northern and Central India. In contrast, infants and young children who continued BF at one and two years, and those who were ever breastfed, had a higher proportion of diarrhoea compared to their counterparts in all regions. The introduction of complementary foods resulted in a high prevalence of diarrhoea among infants aged 6–8 months in the Southern and Central India regions. EIBF and ExcBF were protective against diarrhoea in the North, East and Central Inia regions, as well as at the national level of India. PBF, BotF, continued BF at one year and the introduction of complementary foods were predictors of diarrhoea in Central India and at the national level of India.

Empirical evidence suggests that optimal breastfeeding practices are protective against diarrhoeal disease in populations [1,40]. Our study demonstrated that EIBF was protective against diarrhoea in the Northern, Southern, Eastern, Central, and North-Eastern India regions, as well as at the national level in India. Similarly, ExcBF was protective against diarrhoea in Northern, Eastern and Central India regions, as well as at the national level in India. Our findings were consistent with studies conducted in Andhra Pradesh [41], West Bengal [32], Lucknow [42] and Pudicherry [43] in India, and studies from Nepal [44], Bangladesh [45,46], Nigeria [30,47], Tanzania [29] and Vietnam [48]. These studies found that EIBF and ExcBF were protective against diarrhoea. Our findings have important policy implications for current and/or future efforts to reduce the diarrhoeal disease burden in India. The evidence suggests that current Indian breastfeeding programs, such as the Mother’s Absolute Affection programme [49], which aims to support, generate and provide an enabling environment for mothers, family members and community, need to also consider informing mothers and their families of the additional benefits of optimal breastfeeding, including diarrhoeal disease prevention. 

Additionally, the reassessment and full implementation of global initiatives such as the International Code of Marketing of Breastmilk substitutes (hereafter referred to as The Code [50] and the Baby Friendly Hospital Initiative (BFHI) [51] are also warranted to increase optimal breastfeeding in regional India [4]. The 2016 India Breast Milk Substitute report indicated that The Code is legally enforced in India; however, the effectiveness of this regulatory framework is often not evident at the subnational and national levels [52]. The BFHI indicates that all pregnant women and their families should be informed of the importance of breastfeeding [53]. Although the BFHI has been instrumental in improving breastfeeding in many communities in Southern India [54,55,56], the India’s BFHI is yet to be streamlined for better implementation and benefits. The 2015–2018 World Breastfeeding Trends Initiative (WBTi) for India scored the country zero on BFHI, as the country has no designated “baby friendly” maternity centres (public and private) [57,58]. A lack of BFHI-certified maternity facilities may have considerable implications for appropriate breastfeeding in India, with subsequent impacts on diarrhoeal burden in regional India.

Our study found that PBF was associated with diarrhoea in Central India and at the national level in India. This finding was consistent with past studies conducted in Vietnam [48], Sub-Saharan African countries [28] and Tanzania [29] which showed that PBF increased the likelihood of infants to experience diarrhoea. A possible reason for this association may be due to the inadequate access to sanitary environment and a possible lack of clean potable water for infants in those environments, as PBF also involves the provision of water, water-based drinks and fruit juice to infants aged 0–5 months. In contrast, a multicentre cohort study conducted in India, Ghana and Peru [59], as well as past studies conducted in Nigeria [30] and Bangladesh [46], showed that PBF was protective against diarrhoea. Similarly, previous studies have demonstrated that PBF was not only protective against diarrhoea but also associated with higher intelligence [60] and increased level of education among children [60,61], as well as better economic advancement in later life [61]. The WHO indicates that the intake of water, tea, honey and other non-nutritive fluids in addition to breast milk increases the pathogenic contamination of these fluids which may cause the infant to experience diarrhoea [62]. In India, where access to potable water and sanitation and hygiene remains a significant public health challenge, advocating for PBF behaviour in many Indian communities may likely increase the burden of diarrhoeal disease among children. The Indian government has recently implemented new policies, such as the Swacch Bharat Abhiyan [63], to increase access to safe drinking water, prevent open defecation and maintain environmental sanitation. The integration of these WASH programmes with breastfeeding interventions will maximise the impacts of infant and young-child feeding policy interventions in the short- and long-term in India.

BotF is one of the key IYCF indicators, as it can play a major role in child health [31]. In the present study, infants and young children who were bottle fed were more likely to experience diarrhoea in the Northern, Eastern and Central India regions, as well as at the national level in India. Our results have been documented in studies conducted in Pakistan [64], the Philippines [65] and Nigeria [30]. A study conducted in rural Punjab [66] and another hospital-based cross-sectional study from India [67] revealed that Rotavirus infection (the most common diarrhoea-causing organism) was positive in children who were bottle fed. This is possibly due to a lack of sanitary and hygienic practices in maintaining the bottles and the preparation of the food. The Breastfeeding Promotion Network of India (BPNI) has provided recommendations against the use of BotF and has also refrained from taking any funds from baby-food-manufacturing companies, in order to fully advocate for appropriate IYCF practices in India [58]. While these efforts are commendable, more regional infant feeding organisations need to introduce and implement similar efforts to reduce the use of BotF in India.

Our study showed that the introduction of complementary foods to infants aged 6–8 months was associated with diarrhoea in Central India. This finding was consistent with previous studies conducted at the national level in India [68,69], where the introduction of complementary foods was associated with diarrhoea. Studies conducted in rural Punjab [66] and West Bengal [35] also suggested that the introduction of complementary foods was associated with diarrhoea among infants. Similarly, past studies conducted in Tanzania [29] and Nigeria [30] demonstrated that the introduction of complementary foods in infants aged 6–8 months was associated with an increased likelihood of infants to experience diarrhoea. These results may likely be due to possible contamination of the infant food, poor hygiene and inadequate sanitary storage practices/facilities [70,71,72]. The present study also found that children aged 20–23 months whose mothers continued BF at two years of age were more likely to experience diarrhoea compared to their counterparts in Western India. A possible reason for this finding may be due to the concurrent introduction of contaminated complementary foods [69].

In India, various factors influence the choice of complementary foods, including socioeconomic status, mother’s education, cultural beliefs, and regional variations in infant foods availability and accessibility [5]. Female autonomy and the presence of a key family member [73] and affordability of local foods [74,75] are some of the other factors that determine the choice of infant complementary foods. A single one and/or combination of these factors are likely to also contribute to the diarrhoeal disease burden in regional India. Therefore, adequate training for health professionals on evidence-based interventions to improve infant feeding practices (such as complementary food preparation, safe handling and hygienic storage) is essential to reduce the diarrhoeal disease burden in Indian communities. Recently, the Indian government has been supporting essential maternal and child health interventions, including the Stop Diarrhoea Project (SDP) [76]. While these initiatives are essential in many Indian communities, it is imperative that they are integrated with IYCF initiatives and should be culturally appropriate and context-specific, to maximise impacts.

### Study Limitations and Strengths 

Relevant methodological limitations need to be considered when interpreting the study findings. Firstly, recall bias may have affected the results, as data were collected through self-report. We, however, made efforts to reduce the potential impact of recall bias, as the analyses were limited to the youngest child who lived with the respondent. Secondly, there may have been misclassification bias, as diarrhoea measurement was based on the two weeks prior to the survey. That is, respondents may have incorrectly indicated that their infants or young children had diarrhoea, when it may have just been a minimal change in the bowel habit at that moment in time. This may have led to an overestimation or underestimation of the measure of association between the exposures and outcome. Thirdly, our inability to account for all the confounding factors (e.g., food safety or cultural variations in complementary foods) may have impacted the association between IYCF practices and diarrhoea. Fourthly, it is difficult to establish a clear temporal association between IYCF practices and diarrhoea due to the cross-sectional data employed in the study. Despite these limitations, a major strength of our study includes the use of the most recent NFHS-4 data, which provides up-to-date information on India’s IYCF practices and diarrhoea. Moreover, the study findings are less likely to be influenced by selection bias, as the response rate in the survey was high, obtained by the use of standardised data-collection methods [24].

## 5. Conclusions

The present study showed that EIBF and ExcBF were protective against diarrhoea in regional India, while PBF, BotF, continued BF at two years and ISSSF were risk factors for diarrhoea. There is a need for an integrated and a multilevel approach for strategic policy implementation in all the regions of the country, to ensure that there are considerable improvements in IYCF behaviours with the subsequent impact on diarrhoeal disease in Indian children. 

## Figures and Tables

**Figure 1 ijerph-17-04740-f001:**
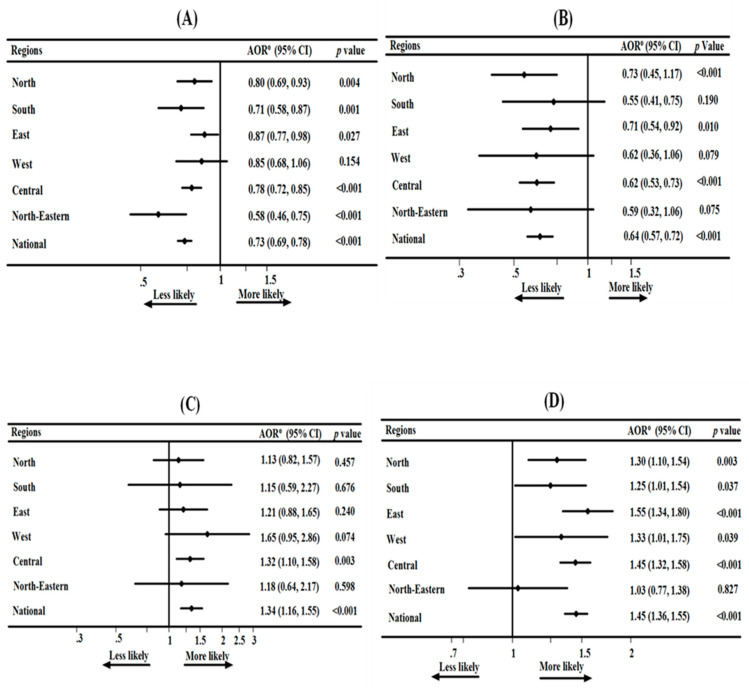
(**A**) Association between early initiation of breastfeeding and diarrhoea in regional India; (**B**) association between exclusive breastfeeding and diarrhoea in regional India; (**C**) association between predominant breastfeeding and diarrhoea in regional India; (**D**) association between bottle feeding and diarrhoea in regional India; (**E**) association between continued breastfeeding at one year and diarrhoea in regional India; (**F**) association between continued breastfeeding at two years and diarrhoea in regional India; (**G**) association between children ever breastfed and diarrhoea in regional India; (**H**) association between introduction of solid, semi-solid or soft foods and diarrhoea in regional India; 95% CI = 95% confidence interval; AOR = adjusted odds ratio; models adjusted for child, maternal, family, media, health service, environmental and community-level factors.

**Table 1 ijerph-17-04740-t001:** Prevalence and 95% confidence intervals (CIs) of diarrhoea among children aged 0–23 months by infants and young child feeding practices in regional India (N = 90,596).

	North		South		East		West		Central		North-East	
	*N ^a^*	Prevalence ^b^ (95% CI)	*N*	Prevalence (95% CI)	*N*	Prevalence (95% CI)	*N*	Prevalence (95% CI)	*N*	Prevalence (95% CI)	*N*	Prevalence (95% CI)
**Early initiation of breastfeeding**												
No	939	8.3 (7.7, 9.1)	860	5.2 (4.6, 5.9)	1770	7.5 (7.1, 8.1)	749	6.5 (5.7, 7.4)	3672	14.7 (14.2, 15.3)	82	2.5 (2.1, 3.1)
Yes	393	3.5 (3.1, 3.9)	699	4.2 (3.7, 4.7)	1163	4.9 (4.5, 5.4)	797	6.9 (6.0, 8.0)	1298	5.2 (4.9, 5.5)	85	2.6 (2.3, 3.0)
**Exclusive breastfeeding**												
No	169	6.1 (5.1, 7.3)	114	3.1 (2.2, 4.2)	260	5.0 (4.2, 5.9)	126	4.9 (3.4, 6.9)	638	10.3 (9.4, 11.1)	13	1.7 (1.2, 2.3)
Yes	132	4.8 (3.9, 5.8)	120	3.2 (2.4, 4.4)	241	4.6 (3.9, 5.4)	118	4.6 (3.5, 6.0)	412	6.6 (6.0, 7.4)	13	1.6 (1.0, 2.4)
**Predominant breastfeeding**												
No	226	8.2 (7.1, 9.5)	198	5.4 (4.3, 6.8)	404	7.7 (6.8, 8.8)	178	6.9 (5.3, 8.8)	742	11.9 (11.1, 12.9)	22	2.7 (2.0, 3.7)
Yes	75	2.7 (2.1, 3.5)	36	0.9 (0.5, 1.6)	98	1.8 (1.4, 2.4)	67	2.6 (1.6, 4.1)	309	5.0 (4.4, 5.6)	4	0.5 (0.3, 0.8)
**Bottle feeding**												
No	91	8.8 (8.2, 9.5)	1133	6.8 (6.2, 7.2)	2339	10.0 (9.4, 10.6)	1260	10.9 (9.9, 12.0)	3584	14.4 (13.9, 14.9)	137	4.2 (3.7, 4.8)
Yes	342	3.1 (2.6, 3.5)	426	2.5 (2.2, 3.0)	595	2.5 (2.2, 2.8)	287	2.5 (1.9, 3.2)	1385	5.6 (5.2, 5.9)	29	0.9 (0.7, 1.1)
**Continued breastfeeding at one year**												
No	35	1.8 (1.2, 2.8)	80	2.7 (1.8, 5.3)	43	1.1 (0.7, 1.5)	47	2.3 (1.2, 4.4)	121	3.0 (2.4, 3.6)	2	0.3 (0.1, 0.5)
Yes	212	11.1 (9.8, 12.9)	195	6.7 (5.3, 8.4)	526	12.8 (11.5, 14.3)	258	13.0 (10.7, 15.7)	733	18.0 (16.7, 19.3)	32	5.2 (4.1, 6.7)
**Continued breastfeeding at two years**												
No	38	2.3 (1.6, 3.3)	106	4.0 (2.9, 5.3)	72	1.9 (1.3, 2.6)	38	2.1 (1.3, 3.4)	159	4.5 (3.8, 5.2)	7	1.3 (0.8, 3.0)
Yes	127	7.8 (6.4, 9.4)	151	5.8 (4.3, 7.4)	349	9.2 (8.0, 10.6)	144	7.9 (5.9, 10.4)	471	13.2 (12.1, 14.5)	21	4.1 (3.0, 5.5)
**Children ever breastfed**												
No	36	0.3 (0.2, 0.4)	76	0.5 (0.3, 0.7)	58	0.2 (0.2, 0.4)	31	0.3 (0.1, 0.4)	125	0.5 (0.4, 0.6)	9	0.3 (0.1, 0.6)
Yes	1297	11.5 (10.8, 12.3)	1483	9.0 (8.2, 9.8)	2876	12.3 (11.7, 13.0)	1516	13.2 (11.9, 14.5)	4845	19.5 (18.9, 20.1)	157	4.8 (4.3, 5.5)
**Introduction of solid, semi-solid or soft foods**												
No	137	8.2 (6.9, 9.7)	93	4.2 (3.1, 5.7)	281	8.5 (7.2, 9.8)	171	10.5 (8.1, 13.6)	451	12.5 (11.3, 13.8)	11	2.9 (1.9, 4.6)
Yes	103	6.1 (5.1, 7.5)	170	7.7 (5.5, 10.6)	183	5.5 (4.5, 6.6)	151	9.3 (6.6, 12.9)	355	9.9 (8.8, 11.0)	11	3.1 (2.0, 4.8)

*N****^a^*** = weighted total number of children aged 0–23 months within each IYCF indicators; 95% CI: 95% confidence interval. Prevalence **^b^** = represents the overall weighted proportion of children with diarrhoea for each level (“No” or “Yes”) of infant and young-child feeding indicators. Early initiation of breastfeeding was defined as the proportion of children within 0–23 months of age who were breastfed within one hour of birth. Exclusive breastfeeding was defined as the proportion of infants 0–5 months of age who received breast milk as the only source of nourishment but allowed oral rehydration solution, drops or syrups of vitamins and medicines. Predominant breastfeeding was defined as the proportion of infants 0–5 months of age who received breast milk as the main source of nourishment but allowed water, water-based drinks, fruit juice, oral rehydration solution, drops or syrups of vitamins and medicines. Bottle feeding was defined as the proportion of children 0–23 months of age who were fed with a bottle during the previous day. Continued breastfeeding at one year was defined as the proportion of children 12–15 months of age who were fed breast milk. Continued breastfeeding at 2 years was defined as the proportion of children 20–23 months of age who were fed breast milk. Children ever breastfed was defined as the proportion of children born in the last 24 months who were ever breastfed. Introduction of solid, semi-solid or soft foods was defined as the proportion of infants 6–8 months of age who received solid, semi-solid or soft foods.

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
