# Peer review of "Regional Analysis of Associations between Infant and Young Child Feeding Practices and Diarrhoea in Indian Children"

_ijerph, 2020, doi:10.3390/ijerph17134740_

Round 1

Reviewer 1 Report

Regional analysis of associations between infant and young child feeding practices and diarrhoea in Indian children

The aim of this study was to examine the relationships between the infant and young child feeding practices and diarrhoea in different regions of the India. The sample is large and the variables related nutrition were well chosen. In general, I consider that the study is very appropriate.  

Nevertheless, I think the results need to be written clearer. It is hard to follow what the authors are trying to say, and what are the main messages they want the reader to take away. There are a lot of outcomes and a lot of analyses, being difficult to found conclusions. All good work, but needs to be clearer.

As minor aspects:

Did the subjects have to complete all the tests and data collection to be included in the study? minimum percentage?

Approximately how many covariates were included in the final models?

Author Response

Thank you

Reviewer 2 Report

The association between young child feeding practices and diarrhea was important topic, especially in developing countries. A body of studies has focused on it, and significant association has been found. However, variation and special characteristics could exist with time and across countries about such association of interest. So it is important to understand well concrete association, like India, which will be helpful for maternal and child health care in India. However, some issue should be addressed further. 1. About covariate adjustment, authors provided the process of staged modeling technique. It would be better if they could provide reference about this technique. 2. In the results on adjustment, authors did not provide clearly adjusted-covariates in the figure 1, also they should provide statement on adjustment of clustering and sampling weights in figure 1. 3. About data at national level, how did authors combine the data from regions? 4. How did author select the subjects from the national data set? whether or not there were some missing data in this study?

Author Response

Thank you.

Round 2

Reviewer 1 Report

Thank you for considering my comments

Author Response

Thank you.

Reviewer 2 Report

Authors have addressed most of my comments, and the manuscript has been improved much. However, the following question should be addressed before publication.

When author select the subjects from the national data set, whether or not there were some missing data in this study? 

Author Response

Thank you.
